# HIV incidence and risk contributing factors among men who have sex with men in Benin: A prospective cohort study

Septime P. H. Hessou[1,2,3☯*], Yolaine Glele-Ahanhanzo[4☯], Rhéda Adekpedjou[5], Clement Ahoussinou[6‡], Codjo D. Djade[2‡], Alphonse Biaou[7‡], Christian R. Johnson[3‡], Michel Boko[3‡], Michel Alary[2‡]

**1** National Reference Centre for AIDS Research and Care (CNRRPEC-CNHU/Bénin), Cotonou, Bénin, **2** Population Health and Best Practices in Health Axis Quebec City University Hospital Research Center Laval University, Hôpital du Saint-Sacrement, Québec, Canada, **3** Inter-faculty Centre for Training and Research in Environment for Development (CIFRED), Abomey-Calavi University (UAC), Calavi, Bénin, **4** Regional Institute of Public Health (IRSP), University of Abomey-Calavi (UAC), Ouidah, Bénin, **5** Centre de Recherche du Centre Hospitalier de l'Université de Montréal (CRCHUM), Université de Montréal, Quebec, Canada, **6** Institute for the Analysis of Communication and Social Groups, Missereté, Bénin, **7** National Program against Buruli Ulcer, Cotonou, Benin

☯ These authors contributed equally to this work.
‡ These authors also contributed equally to this work.
* shessou@yahoo.fr

**Data Availability Statement:** If the data are all contained within the manuscript and/or Supporting Information files, enter the following: All relevant

## Abstract

Men who have sex with Men (MSM) are a key population in the transmission of Human Immunodeficiency Virus (HIV) infection. In Benin, there is a lack of strategic information to offer appropriate interventions for these populations who live hidden due to their stigmatization and discrimination. The objective is to identify contributing factors that affect HIV incidence in the MSM population. Study of a prospective cohort of 358 HIV-negative MSM, aged 18 years and over, reporting having had at least one oral or anal relationship with another man during the last 12 months, prior to recruitment. The monitoring lasted 30 months with a follow-up visit every six months. Univariate analyses and a Cox proportional hazards multivariate regression were used to examine the association between bio-behavioral, socio-demographic and knowledge-related characteristics with HIV incidence. The retention rate for the follow-up of the 358 participants was 94.5%. On the 813.5 person-years of follow-up, 48 seroconversions with an HIV incidence of 5.91 per 100 person-years were observed (95% CI: 4.46–7.85). Factors associated with the high risk of HIV were age (HR = 0.4; 95% CI: 0.2–0.8), living in couple (HR = 0.5 95% CI: 0.2–0.96) and the lack of condom systematic use with a male partner during high-risk sex (HR = 3.9; 95% CI: 1.4–11.1). HIV incidence is high within MSM population and particularly among young people. Targeted, suitable and cost-effective interventions for the delivery of the combination prevention package in an environment free of stigma and discrimination are necessary and vital for reaching the 90x90x90 target.

data are within the manuscript and its Supporting Information files.

**Funding:** The authors received no specific funding for this work.

**Competing interests:** The authors have declared that no competing interests exist.

## Introduction

Homosexuality in most sub-Saharan African countries is a strong source of stigma, discrimination and rejection [1, 2]. Most MSM are forced to hide and cut themselves off from others to live their homosexuality [2, 3]. Men who have sex with Men (MSM) have been largely affected by HIV since the beginning of the epidemic [4–6]. In France, the latest epidemiological data on HIV infection depict a worrying picture: sex between men is the only mode of contamination for which there has been no decrease in new infections since the beginning of this decade. Over the same period, among MSM, an increase in sexual risk behaviors has been observed, as well as an increase in sexually transmitted infections, including syphilis [7]. It should be noted that HIV prevalence remains high within MSM populations in most sub-Saharan African countries. It ranged from 5.6% (95% CI: 2.3–9.8) to 49.5% (95% CI: 42.5%-56.5% in the following countries: Agadir in Morocco, Johannesburg in South Africa, Togo, Malawi, Uganda, Cameroon, Lagos in Nigeria, and Senegal [4, 8–14]. Prevalence of HIV infection among MSM populations in these countries are 5 to 15 times higher than the national prevalence in the general population [15–17].

Very few studies have been conducted in Africa to really estimate the incidence of HIV infection. Mathematical and statistical modellings, through different studies on the transmission modes and the weight of HIV epidemic in key populations, have provided data to meet intervention planning needs for this target [15, 18]. This is justified by difficulties met in following a cohort of HIV-free subjects within the general population, sometimes at low risk and for a reasonable period of time, so as to provide a reliable estimate of the incidence of this disease. The advent of monitoring populations at higher risk of HIV infection: key populations, especially MSM and their self-sufficient lifestyles have made prospective cohort studies, possible today.

It should be noted that a better knowledge of HIV-infection incidence and its evolution over time are the parameters required to improve the quality of resource planning and capacity to manage patients [1, 15, 18, 19]. The objective of the study is then to identify the factors that contribute to the incidence of HIV infection in a population of MSM in Benin.

## Materials and methods

### Type of study, populations and sampling

This is a prospective cohort study. HIV-negative MSM were recruited from a first seroprevalence study conducted from January 2016 to April 2016 in Benin. Recruitment began on April 15, 2016 and ended on May 30, 2016. Monitoring began with the inclusion of the last subject. It started on June 2016 and ended on November 2018. It lasted 30 months.

### Sampling

As a result of the baseline seroprevalence survey, the number of MSM that were initially included in the follow-up cohort is equal to the number of HIV-negative MSM (n = 358 MSM). The Schwartz equation was used to determine the sample size of the baseline survey and the RDS method was used for the enrolment of MSM[1]. The size was 414 MSM participants at the end of this first study; 56 MSM participants were tested positive. All MSM tested negative at the end of this baseline study, that is 358 people, agreed to take part in the prospective cohort [20].

### Inclusion criteria

Inclusion in the cohort was proposed to all eligible subjects, that is the 358 HIV-negative MSM from the baseline study. Inclusion was made on a voluntary basis, after giving detailed

information to each participant and obtaining their written and informed consent. Note that to be eligible for the baseline study, one must be a biological male, at least 18 years of age, and have had at least one instance of anal (receptive or insertive) or oral (fellatio and/or anulingus) sex with a male partner in the 12 months prior to the survey.

## Cohort follow-up

On the basis of a monitoring sheet, three types of parameters were analyzed during the longitudinal follow-up of MSM: (i) risk behaviors, (ii) clinical examination data and (iii) biological data (HIV and other STIs screening).

Each of the 6 follow-up teams consists of an investigator who is familiar with the MSM environment, a laboratory technician and a doctor. The capacities of the doctor and laboratory technician on health issues related to MSM have been strengthened beforehand on the management of MSM. This follow-up team submitted MSM participants to 01 follow-up visit over a three-week period per team, every six months and each MSM is allowed to come as many times as he wishes during the follow-up period (Fig 1).

## Monitoring organization

This is a closed cohort. Monitoring took place every 6 months and started at the same time for the 358 subjects initially eligible for the 5 consecutive semesters that is 30 months of follow-up. A monitoring visit was organized every last month of the semester. It took three weeks per team, so as to allow each participant to plan and make himself available for the follow-up session.

During the follow-up, the review focused on the MSM sexual health. It involved the whole body and more specifically the mouth, the genital tract, the perineal and anorectal areas. The follow-up was carried out on secure and confidential sites (set up and equipped for the purpose and participants were informed one month in advance by MSM peer interviewers).

The end of the follow-up takes place when the participant becomes HIV positive, that's when he/she shows HIV positive.

The participant who has been lost from sight (the lost from sight participant) is the one who was absent at the first follow-up (lost from sight noticed, and still absent at the second follow-up: lost from sight confirmed at the second: he is lost from sight).

## Variables collected during monitoring

- Socio-demographic variables: age, marital status, occupation, type of homosexuality, level of education, sexual orientation at the beginning of the monitoring.

- The variables collected during the follow-up: the number of casual sexual partners, condom tearing, type of lubricant used, frequency use of water-based lubricant, signs of sexually transmitted infection, condom use with lubricant gel during anal sex with a male partner (last 3 months, at last anal intercourse, and at each intercourse), knowledge of partner's serological status, knowledge of HIV infection, follow-up time and occurrence of HIV infection were collected every 6 months.

## Data collection tools and techniques

Behavioral data were collected from a questionnaire. At the end of the administration of the questionnaire, and after pre-test counseling, respondents were submitted to a screening test

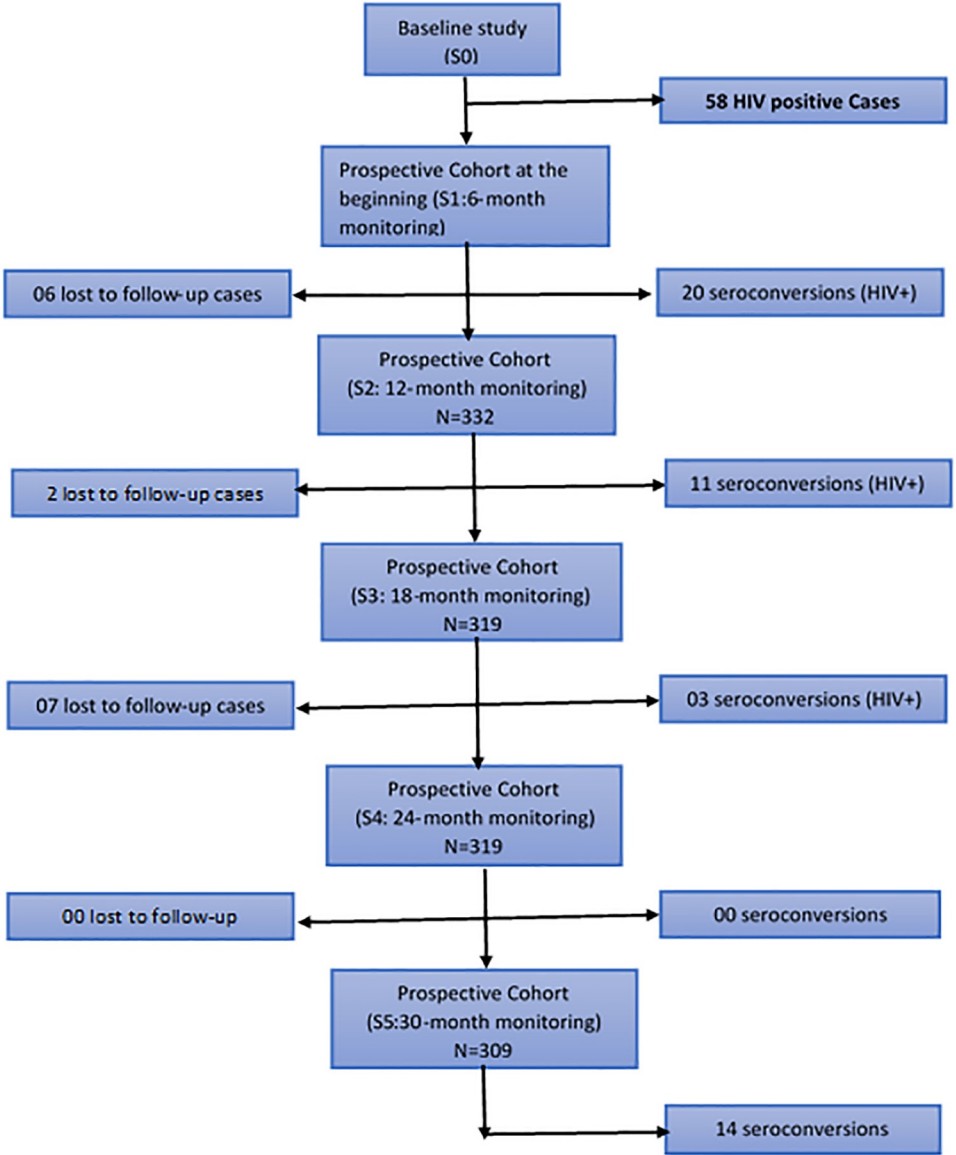

**Fig 1. Follow-up chart of men who have sex with Men at the prospective cohort for 30 months for estimation of HIV infection, Benin 2018.**

followed by post-test counseling and the result for the biological component. A synthesis of all the data collected was carried out every six months according to the study protocol.

Arrangements have been made to notify on time MSM and raise awareness about each data collection period with the help and support of peer outreach educators. For the search of those "lost to follow-up", an active search system has been set up to limit their number as much as possible. Thus, during their enrolment, their various contacts (telephone contact, e-mail, contact of the sponsor and the phone number of a contact person) were registered to facilitate their search. A lump sum comprised between US$5 and US$10, depending on the distance from the place of residence was retained, according to the national rates, to cover the participants' transport costs.

## Data processing and analysis

Data were entered using an input mask, with the Epi Data software, followed by a clearance and data consistency checks.

At the descriptive level, HIV incidence was estimated by dividing the number of new HIV cases during the reference period by the number of person-years of time. Data from each half-yearly monitoring were combined with the data of the preceding half-year. Thus, the final incidence rate will be calculated at the end of the last monitoring appointment.

From an analytical standpoint, data analysis was based on the modeling of HIV-infection survival. The dependent variable considered here is twofold (HIV seropositivity; follow-up time). A survival model was developed using the Kaplan-Meier statistical method with a survival curve of HIV infection over time. The Cox proportional hazards regression was performed to measure the association between socio-demographic, behavioral characteristics and HIV knowledge and incidence. Variables issued from the univariate analysis were included in the Cox multiple stepwise regression model with predictor of $P < 0.2$. The final model kept the significant variables at a threshold of 0.05. This made it possible to determine the Adjusted Hazard Risk Ratio (aHR) for factors related to HIV seroconversion in order to identify the contributing factors to HIV infection among MSM. The assumption on proportional risk was verified by the Schoenfeld Residuals Testing.

**Ethical and deontological considerations.** The protocol for this prospective cohort study has been approved by the Comité National d'Ethique pour la Recherche en Santé du Bénin (National Ethics Committee for Health Research in Benin: CNERS). MSM in the study received periodic information on the study and monitoring conditions and gave, each time, in the light of these information, their informed and written consent. At each visit, they received a travel fee less than or equal to US$10, as well as lubricating gel condoms and free STI care.

## Results

### Socio-demographic factors

The cohort study for estimating the incidence of HIV infection lasted 30 months with 5 respective monitoring sessions of 6 months each. 95.81% performed a long-term follow-up (Fig 1), with an average age of 24.34 (3.34) years and a range comprised between 19 and 43 years. In the study, the majority of MSM were single: 91.6%. They are educated with nearly half, 47.8% at a higher level, 32.5% are bisexual and 15.9% have 4 or more sexual partners, 33.8% of MSM were living in couple (Table 1).

### Evolution of behaviors and knowledge

MSM had, on average, between 2.41 (1.49) and 3.13 (1.66) casual sex partners, with a peak in the 3rd quarter between the 12th and 18th months 3.21(.48). The water-based lubricant is the most commonly used by MSM during the monitoring. Its use ranged from 47.8% during the 1st monitoring to 56.3% at the end of the 30-month period of monitoring. The combined use of condoms and lubricant remains around 47% for the different follow-up periods with a drop-out rate of 41.5% in the 3rd quarter between the 12th and 18th months. Knowledge of the partner's serological status at the beginning of the follow-up was 59.0%, it increased to 60.1% at the end of the 30-month period follow-up. Condom tearing during anal sex ranged from 30.4% to 27.5%. (Table 2).

The level of knowledge is low: 32.39% of MSM have a low level of knowledge about HIV infection, 55.2% have an average level and 12.5% have a high level. At 18 months of follow-up, the level of knowledge remained low, at 26%, the average level is 26.9% and the high level is

**Table 1. Number and percent distribution of men who have sex with men (MSM) by socio-demographic characteristics, age, marital status, education, occupation, living in couple, type of homosexuality, sexual practices, number of male partners, Benin 2018.**

| Variables | | Size (N = 358) | % |
|---|---|---|---|
| Ages (years) | | | |
| | Below 25 years | 168 | 46.9 |
| | 25 years and more | 190 | 53.1 |
| Marital status | | n | % |
| | Married/ divorced | 30 | 8.4 |
| | Single | 328 | 91.6 |
| Educational level (N = 358) | | | |
| | Non formal/primary | 39 | 10.9 |
| | Secondary | 148 | 41.3 |
| | Higher than secondary | 171 | 47.8 |
| Occupation | | | |
| | Craftman/workman | 82 | 22.9 |
| | Salesman | 21 | 5.9 |
| | Officer | 38 | 10,6 |
| | Pupils/students | 217 | 60,6 |
| Living in couple | | | |
| | No | 237 | 66.2 |
| | Yes | 121 | 33.8 |
| Type of homosexuality | | | |
| | Homosexual | 225 | 62.8 |
| | Bisexual | 133 | 37,2 |
| Sexual practices | | | |
| | Insertive/active | 198 | 55.3 |
| | Receptive/Passive | 117 | 32.7 |
| | Versatile/ both practices | 43 | 12.0 |
| Number of male partner | | | |
| | 1 | 126 | 35.2 |
| | Between 2 and 3 | 175 | 48.9 |
| | Between 4 and 5 | 57 | 15 |

47.1%. At the end of the monitoring, the level of MSM in HIV infection became high at 52.3%, average at 27.2% and low level at 20.5% (Fig 2).

## Incidence of HIV infection

Study participants had 813.5 person-years of monitoring, with 48 HIV-positive cases resulting in an incidence of 5.91 per 100 person-years (95% CI: 4.46–7.85).

MSM under 25 years of age have an HIV incidence of 8.5 (95% CI: 5.9–12.1) per 100 PY. For those with higher education an incidence of 7.1 (95% CI: 4.5–9.7) per 100 PY has been observed. Married/divorced MSM have an HIV incidence of 7.9 (95% CI: 3.3–19.1) per 100 PY, of 7.7 (95% CI: 5.1–11.7) among bisexual MSM. MSM with more than 3 sexual partners have an HIV incidence of 6.6 (95% CI: 4.23–9.1) per 100 PY. Those who do not always use both condoms and gel have an HIV incidence of 7.9 (95% CI: 5.8–10.8) per 100 PY. MSM who do not consistently use condoms during anal sex with a male partner have an incidence of 6.4 (95% CI: 3.6–9.2) per 100 PY. As for MSM who do not know their HIV status, they have an incidence of 7.5 (95% CI: 5.0–11.5) per 100 PY.

**Table 2. Summary of HIV-infection risk factors among men who have sex with men during a 5-semester (30-month) monitoring period, Benin 2018.**

| Characteristics | Modalities | Monitoring (months) | | | | |
|---|---|---|---|---|---|---|
| | | 6 | 12 | 18 | 24 | 30 |
| Casual sex partners | Average (STD) | 2.41(±1.49) | 3.15(±1.53) | 3.21(±1.48) | 2.28(±1.34) | 3.13(±1.62) |
| Condom tearing | | n = 358 | n = 332 | n = 319 | n = 309 | n = 309 |
| | Oui | 109 (30.4) | 99 (29.8) | 129 (40.4) | 116(37.5) | 86 (27.8) |
| Type of lubricant used n (%) | | n = 358 | n = 330 | n = 319 | n = 309 | n = 309 |
| | Saliva | 118 (33.0) | 107 (32.4) | 91 (28.5) | 81 (26.2) | 71 (23) |
| | vaseline/butter shea | 46 (12.8) | 43 (13) | 42 (13.2) | 41 (13.3) | 41 (13.3) |
| | Water | 171 (47.8) | 158 (47.9) | 163 (51.1) | 164 (53.1) | 174 (56.3) |
| | Body milk/lotion | 23 (6.4) | 22 (6.7) | 23 (7.2) | 23 (7.4) | 23 (7.4) |
| Frequency use of water-based lubricant | | n = 358 | n = 332 | n = 319 | n = 309 | n = 309 |
| | Always | 144 (40.2) | 144 (43.4) | 112 (35.1) | 126 (40.8) | 129 (41.7) |
| | Sometimes | 169 (47.2) | 152 (45.8) | 161 (50.5) | 148 (47.9) | 153 (49.5) |
| | Never | 45 (12.6) | 36 (10.8) | 46 (14.4) | 35 (11.3) | 27 (8.7) |
| Frequency use of condom+ lubricant | | n = 358 | n = 328 | n = 319 | n = 309 | n = 308 |
| | Always both | 125 (34.9) | 150 (45.7) | 133 (41.7) | 148 (47.9) | 146 (47.4) |
| | Not always both | 233 (65.1) | 178 (54.3) | 186 (58.3) | 161 (52.1) | 162 (52.6) |
| STD signs during the last three months | | n = 355 | n = 329 | n = 316 | n = 307 | n = 307 |
| | Pain in the genitals | 57 (16.1) | 48 (14.6) | 46 (14.6) | 43 (14.0) | 25 (8.1) |
| | Uretral discharge | 15 (4.2) | 17 (5.2) | 14 (4.4) | 9 (2.9) | 4 (1.3) |
| | Burning during urination | 59 (16.7) | 48 (14.6) | 27 (8.5) | 9 (2.9) | 9 (2.9) |
| | Urinary urgency | 58 (16.4) | 76 (23.1) | 38 (12.0) | 20 (6.5) | 16 (5.2) |
| | Sex itching | 12 (3.4) | 10 (3.0) | 10 (3.2) | 9 (2.9) | 9 (2.9) |
| | Anal bleeding | 12 (3.4) | 10 (3.0) | 21 (6.6) | 22 (7.2) | 23 (7.5) |
| | Genital ulcer/wounds on the sex | 11(3.1) | 11 (3.3) | 3 (0.9) | - | - |
| | Venereal growths/cockscomb | 12 (3.4) | 16 (4.6) | 9 (2.8) | 6 (2.0) | 6 (2.0) |
| Knowledge of the partner HIV-status | | n = 358 | n = 303 | n = 313 | n = 281 | n = 281 |
| | Yes | 229 (64) | 181 (59.7) | 185 (59.1) | 169 (60.1) | 169 (60.1) |

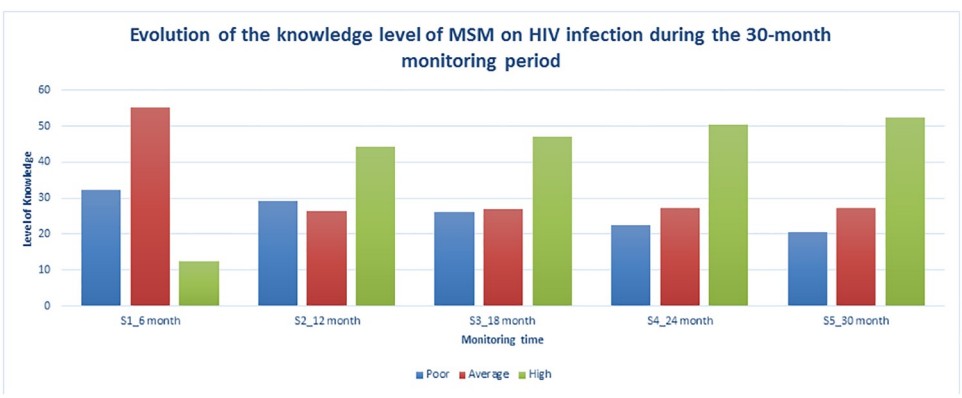

**Fig 2. Evolution of the knowledge level of MSM on HIV infection during the 30-month monitoring period, Bénin 2018.**

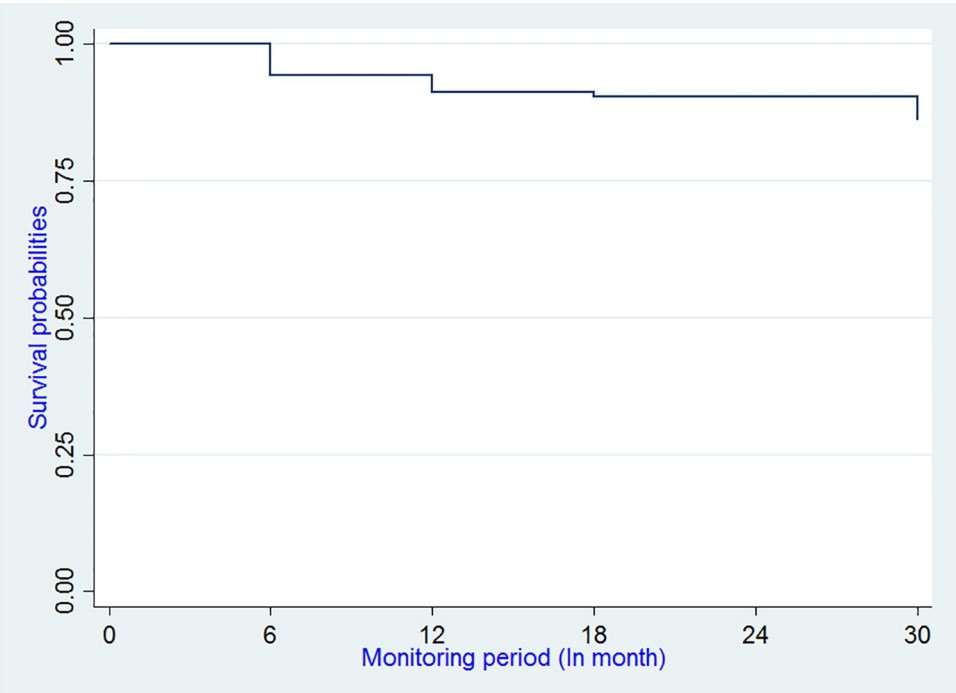

**Fig 3. Kaplan-Meier estimate of the survival function of HIV seroconversion among MSM, Benin 2018.**

Probability of HIV infection varied at 6 months from 94.40% (95% CI: 4.46–7.85) to 91.26% (95% CI: 87.90%–93.77%) at 12 months and 90.40% (95% CI: 86.82–93.04) at 18 months, and to 86.29% (95% CI: 82.22–89.49) at 30 months (Fig 3).

## Factors contributing to HIV seroconversion among MSM

HIV seroconversion risk was lower (HR = 0.4; 95% CI: 0.2–0.8) among MSM aged 25 years and older, compared to those under 25 years of age. MSM who did not live in couple were at lower risk of HIV infection (HR = 0.5; 95% CI: 0.2–0.96) than others. Similarly, MSM who reported using condoms and lubricant during last anal sex with a male partner are at lower risk of HIV infection (HR = 0.4; 95% CI: 0.2–0.9) than others. In contrast, the risk of HIV infection was 3.8 times higher (HR = 3.8; 95% CI: 1.7–8.6) among MSM who reported not always using condoms plus lubricant in the last three months preceding the survey, compared to those who still always used both. Finally, MSM who did not consistently use condoms during anal sex with a male partner were 3.9 times more likely to be HIV-infected than others (HR = 3.9; 95% CI: 1.4–11.1) (Table 3).

## Discussion

This study on the incidence assessment of HIV infection and its associated contributing factors among MSM populations marked a first in Benin's history. It has made it possible to go beyond the usual figures derived from the mathematical modeling of data relating to the study of transmission modes [15, 18]. It has allowed to focus on real field data, so as to measure the HIV spread rate and, consequently, the estimation of new infections cases. It also identified the behavioral risk factors on which actions can be taken to reverse the trend of the epidemic among this MSM population.

**Table 3. Estimated incidence of HIV infection by socio-demographic characteristics and risk factors among men who have sex with men during the 5 semesters of cohort follow-up Benin 2018.**

| Characteristics | Modalities | Monitoring size N(%) | HIV seroconversion N(%) | Person-years PY | Incidence for 100 PY | HR$_b$(CI at 95%) | HRa (CI at 95%) | p value |
|---|---|---|---|---|---|---|---|---|
| | | 358 | 48 | 813.5 | 5.9 | | | |
| Age (years) | | | | | | | | |
| | Less than 25 years | 168(46.9) | 31(64.6) | 365.5 | 8.5(5.9–12.1) | 1 | 1 | |
| | 25 years and over | 190(53.1) | 17(35.4) | 448 | 3.8 (2.3–6.1) | 0.4 (0.2–0.8) | 0.4 (0.2–0.8) | 0.006 |
| Educational Level | | | | | | | | |
| | Higher | 171(47.8) | 27(56.3) | 380.5 | 7.1(4.5–9.7) | 1 | | |
| | Secondary | 148(41.3) | 18(37.5) | 341.5 | 5.3(2.9–7.6) | 0.7(0.4–1.4) | | |
| | Non formal/ primary | 39(10.9) | 3(6.3) | 91.5 | 3.3(-0.4–6.9) | 0.5 (0.1–1.5) | | |
| Occupation | | | | | | | | |
| | Pupils/Students | 217(60.6) | 33(68.8) | 490.5 | 6.7 (4.5–8.9) | 1 | | |
| | Craftsman/worker | 82(22.9) | 10(20.8) | 184.0 | 5.4(2.2–8.7) | 1.2(0.4–3.5) | | |
| | Trader | 21(5.9) | 4(8.3) | 47.5 | 8.4(0.5–16.3) | 0.8 (0.4–1.6) | | |
| | Official | 38(10.6) | 1(2.1) | 91.5 | 1.1(-1.0–3.2) | 0.2(0.0–1.2) | | |
| Marital Status | | | | | | | | |
| | Single | 328(91.6) | 43(89.6) | 750.5 | 5.7 (4.2–7.7) | 1 | | |
| | Married/Divorced | 30(8.4) | 5(10.4) | 63.0 | 7.9 (3.3–19.1) | 0.7(0.3–1.9) | | |
| Living in couple | | | | | | | | |
| | No | 237(66.2) | 38(79.2) | 534.5 | 7.1(5.2–9.8) | 0.5 (0.2–1.0) | 0.5 (0.2–1.0) | 0.040 |
| | Yes | 121(33.8) | 10(20.8) | 279 | 3.6(1.9–6.7) | 1 | 1 | |
| Characteristics | Modalities | Monitoring/ Size N (%) | HIV- positive seroconversion N (%) | Person-years PY | Incidence for 100 PY | RR$_b$(CI at 95%) | RRa (CI) | p value |
| Homosexuality type | | | | | | | | |
| | Homosexual | 225 (62.8) | 26 (54.2) | 527.5 | 4.9 (3.3–7.2) | 1 | | |
| | Bisexual | 133 (37.2) | 22 (45.8) | 286.0 | 7.7 (5.1–11.7) | 1.5 (0.9–2.7) | | |
| Sex practices | | | | | | | | |
| | Insertive/active | 198 (55.3) | 23 (47.9) | 465.5 | 4.9 (3.3–7.4) | 1 | | |
| | Receptive/insertive | 117 (32.7) | 18 (37.5) | 254.5 | 7.1 (4.4–11.2) | 1.4 (0.7–2.6) | | |
| | Versatile/ both | 43 (12.0) | 7 (14.6) | 93.5 | 7.5 (2.2–12.8) | 1.5 (0.6–3.4) | | |
| Number of male sex partners | | | | | | | | |
| | 1 | 126 (35.2) | 14 (29.2) | 293.5 | 4.8 (2.3–7.2) | 1 | | |
| | 2–3 | 175 (48.9) | 26 (54.2) | 394.0 | 6.6 (4.2–9.1) | 1.4 (0.7–2.6) | | |
| | 4–5 | 57 (15.9) | 8 (16.7) | 126.0 | 6.4 (2.1–10.6) | 1.3 (0.5–3.1) | | |
| Condom tearing while having sex during the last three months | | | | | | | | |
| | Yes | 109 (30.4) | 17 (35.4) | 238.5 | 7.13 (3.9–10.4) | 1 | | |

(*Continued*)

**Table 3.** (*Continued*)

| | | | | | | | | |
|---|---|---|---|---|---|---|---|---|
| | No | 249 (69.6) | 31 (64.6) | 575 | 5.4 (3.6–7.2) | 1.3 (0.7–2.2) | | |
| Type of lubricant used | | | | | | | | |
| | Water-based lubricant | 171 (47.8) | 21 (43.8) | 394 | 5.3 (3.1–7.6) | 1 | | |
| | Saliva | 118 (33.0) | 20 (41.7) | 260 | 7.7 (4.5–10.9) | 0.7 (0.4–1.3) | | |
| | Vaseline (petroleum jelly) Ointment | 46 (12.8) | 4 (8.3) | 105.5 | 4.0 (0.2–7.4) | 1.4 (0.5–3.9) | | |
| | Body milk/hand lotion | 23 (6.4) | 3 (6.3) | 54 | 5.6 (0.1–7.6) | 0.9 (0.3–2.9) | | |
| Frequency use of lubricant during the last three months | | | | | | | | |
| | Always | 144 (40.2) | 14 (29.2) | 332 | 4.2 (2.5–7.1) | 1 | | |
| | Sometimes | 169 (47.2) | 27 (56.3) | 377.5 | 7.1 (4.9–10.4) | 1.7 (0.9–3.2) | | |
| | Never | 45 (12.6) | 7 (14.6) | 104 | 6.7 (3.2–14.1) | 1.6 (0.6–3.9) | | |
| Frequency use of condoms and lubricants during the last three months | | | | | | | | |
| | Always both | 125 (34.9) | 7 (14.6) | 298 | 2.3 (1.1–4.9) | 1 | 1 | |
| | Not always both | 233 (65.1) | 41 (85.4) | 515.5 | 7.9 (5.8–10.8) | 3.3 (1.5–7.4) | 3.8 (1.7–8.6) | 0.001 |
| Condom use during the last intercourse with a male partner | | | | | | | | |
| | Yes | 297 (83.0) | 43 (89.6) | 670.5 | 6.4 (4.6–8.3) | 1 | | |
| | No | 61 (17.0) | 5 (10.4) | 143 | 3.5 (0.5–6.5) | 0.5 (0.2–1.4) | | |
| Condom use together with a lubricant during the last anal intercourse with a male partner | | | | | | | | |
| | Yes | 40 (11.2) | 10 (20.8) | 87 | 11 (6.2–21.3) | 1 | | |
| | No | 318 (88.8) | 38 (79.2) | 726.5 | 5.2 (3.8–7.2) | 0.5 (0.2–0.9) | 0.45 (0.22–0.91) | 0.026 |
| Condom use at anal sex intercourses with a male partner during the last three months | | | | | | | | |
| | Yes | 218 (60.9) | 28 (58.3) | 500.5 | 5.6 (3.6–7.6) | 1 | | |
| | No | 140 (39.1) | 20 (41.7) | 313 | 6.4 (3.7–9.1) | 1.1 (0.6–2.0) | | |
| Use of condom together with a lubricant gel at anal intercourses with a male partner during the last three months | | | | | | | | |
| | Yes | 90 (25.1) | 4 (8.3) | 216 | 1.9 (0.1–3.7) | 1 | | |
| | No | 268 (74.9) | 44 (91.7) | 597.5 | 7.4 (5.3–9.5) | 3.9 (1.4–10.8) | | |
| Consistent use of condom during anal sex with a male partner | | | | | | | | |
| | Yes | 226 (63.1) | 29 (60.4) | 516.5 | 5.6 (3.6–7.6) | 1 | 1 | |
| | No | 132 (36.9) | 19 (39.6) | 297 | 6.4 (3.6–9.2) | 1.1 (0.6–2.0) | 3.96 (1.4–11.1) | 0,009 |

(*Continued*)

**Table 3.** (Continued)

| Consistent use of condom together with a lubricant gel during anal sex with a male partner | | | | | | | | |
|---|---|---|---|---|---|---|---|---|
| | Yes | 29 (8.1) | 6 (12.5) | 63 | 9.5 (4.3–21.2) | 1 | | |
| | No | 329 (91.9) | 42 (87.5) | 750.5 | 5.6 (4.1–7.6) | 0.6 (0.2–1.4) | | |
| Took his HIV testing less than 12 months ago | | | | | | | | |
| | Yes | 224 (62.6) | 26 (54.2) | 297.5 | 5.0 (3.4–7.3) | 1 | | |
| | No | 134 (37.4) | 22 (45.8) | 516 | 7.4 (4.9–11.2) | 1.4 (0.8–2.6) | | |
| Is aware of the HV-status of his regular partner | | | | | | | | |
| | Yes | 229 (64.0) | 26 (54.2) | 522 | 5.0 (3.4–7.3) | 1 | | |
| | No | 129 (36.0) | 22 (45.8) | 291.5 | 7.5 (5.0–11.5) | 1.5 (0.8–2.7) | | |
| Complete knowledge of HIV | | | | | | | | |
| | Yes | 158 (44.1) | 22 (45.8) | 350 | 6.3 (3.7–8.8) | 1 | | |
| | No | 200 (55.9) | 26 (54.2) | 463.5 | 5.6 (3.5–7.7) | 1.1 (0.6–1.8) | | |

# Variables with p<0.20 in univariate analysis

## Results' strengths

The prospective cohort study on the estimation of HIV incidence among MSM lasted 30 months with a retention rate of 95.81%. HIV test results are laboratory biological results and not self-reported results. There is near perfect agreement between the results of rapid HIV screening tests performed in the field and the quality assurance results monitored in the reference laboratory (Table 4).

The HIV-infection incidence is high. The following factors: age, living as a couple, condom use and condom and gel use during sexual intercourse have significantly contributed to the occurrence of new HIV infections among MSM populations. It's a call to a new paradigm for designing and implementing interventions in favor of MSM populations in HIV prevention and care.

The level of knowledge of getting HIV infected remained low at least during 18 months.

**Table 4. Summary results of the different levels of concordance between rapid HIV screening tests performed in the field by biomedical laboratory technicians and quality assurance control at the national reference laboratory for each of the 5 phases of cohort follow-up, Benin 2018.**

| Different phases of the study | Cohen's Kappa coefficients | Interpretation of results |
|---|---|---|
| 6 months follow-up | 0.92 | Near-perfect tuning |
| 12 months follow-up | 1.00 | Near-perfect tuning |
| 18 months follow-up | 1.00 | Near-perfect tuning |
| 24 months follow-up | - | |
| 30 months follow-up | 0.95 | Near-perfect tuning |

* interpretation: < 0 = Disagreement; 0.0–0.20 = Very low tuning; 0.21–0.40 = Weak Agreement:

0.41–0.60 = Moderate agreement; 0.61–0.80 = Strong Agreement; 0.81–1.00 = Near-perfect tuning

This is much more related to the lack of effect after the intervention or to an ongoing intervention. The preliminary results of this study have made it possible to correct and adapt intervention strategies in the course of implementation in relation to the specificity of the MSM target. It should be remembered that behavior change is also a process that lasts over time and that progressively gives results.

The frequent tearing of condoms according to MSM would be related to the character and mechanism of the sexual act but especially and above all in the absence of the use of lubricating gels concomitantly with the condom. At the beginning of the study during the first six months of follow-up, just one-third of MSM (34.9%) used the condom coupled with the lubricating gel. The other issue is the availability and accessibility of free lubricant gel as well as condoms in intervention programs.

In this study, the rate of condom tearing is high, this situation according to the MSM would be linked to the character and mechanism of the sexual act but especially and above all in the absence of the use of lubricating gels concomitantly with the condom. At the beginning of the study during the first six months of follow-up, just one-third of MSM (34.9%) used the condom coupled with the lubricating gel. Another issue is the free availability and accessibility of lubricant gel as well as condoms at the intervention program level.

## Comparison: Similarities and differences with other studies

In terms of age distribution, MSM are relatively young in our study, mostly strictly homosexual and with a high level of education. These results are similar to those found in the studies of Jun-Je Xu et al [21] and Dong et al in China [22]. It must be recognized that these are recent studies carried out in a context of global change in the access to new information and communication technologies and in terms of information for the benefit of sexual minorities and especially in favor of the defense of human rights and freedoms. MSM, both in China and Africa, experienced stigma and discrimination; the eldest, who are certainly the most numerous, went into hiding and decided not to participate in studies [23]. This situation is all the more true since a contrary attitude is observed in the study carried out by Tabet et al in the USA where MSM under 25 years of age are 13.7% and only 12.5% have a high level of education [24].

The incidence of HIV infection is similar to that of the prospective cohort study of MSM of Beijing, China, 5.9 per 100 PY (95%CI: 4.6–7.6) [25, 26] with a 32-month monitoring like ours with a retention of 70%. It's also comparable to the HIV incidence of 5.90 per 100 PY (95% CI: 3.6, 9.1%) found among MSM of young black people community in 6 US cities [27]. This result is lower than that of the prospective cohort study conducted among young MSM in China with an HIV incidence of 6.7% per PY. It should be noted that it was a large scale study conducted in several cities, with different levels of HIV infection prevalence and a 12-month monitoring [28]. This incidence of HIV infection is also lower than the one found in the study conducted by Mao et al [29] with 7.83 per 100 PY (95% CI 4.48–12.72) This gap can be explained by the fact that socio-cultural context of studies are different, the 8-month monitoring period is relatively short and targets are oriented towards migrants. Dongliang Li in his study found an incidence of 8.09 per 100 PY (95% CI 6.92–9.26), with a retention of 86.8% of participants over a 12-month monitoring period [30]. This disparity in results is mainly due, not only to the number of MSM enrolled in the initial cohort, but also to the monitoring time and the retention capacity of MSM. Despite all these different parameters noted from one study to another, HIV incidence remains high and calls for specific and customized prevention measures towards MSM and better management for those who are already infected through ARV therapy. Only in this way will we be able to reach 90x90x90 [6, 31].

This study also showed that seroconversion is significantly associated with age, the status of living as a couple with a woman, the use of condoms and especially condoms with lubricating gels. The risk of HIV seroconversion was lower among MSM aged 25 years and older. Dongliang Li et al, in his prospective cohort study on MSM in 2012 in Beijing [30] and Balaji et al, in a study carried out in 2008 in the USA, found similar results [32]. Once again, this demonstrates the sexually active nature of young MSM and their effective contribution to the emergence of new infections, hence the need to consider this group as a priority for HIV prevention interventions. Combined prevention such as pre-exposure prophylaxis (PrEP) would be a promising solution [19, 31, 33].

MSM living in couple with a woman were at higher risk of HIV infection than others. The results of this study are comparable to those found by Beyer et al in South Africa [34], Xu et al in the USA [35] and Solomon et al in India [36, 37]. In the context of MSM, this situation of bisexuality shows that living one's homosexuality and also being married to a woman at the same time is associated with a high probability of being infected with HIV. Indeed, such behavior may lead MSM to have volatile male sex partners, for fear of being discovered, and to maintain a multi-male sexual partnership [36]. It should also be noted that Benin is a mixed epidemic country with high sexual transmission of HIV infection and an estimated HIV concentration among sex workers (SWs) with a prevalence of more than 20% in 2015 [38, 39]. The bisexual nature of a segment of MSM and their close contacts with SWs can be a double burden of HIV infection and would explain this situation. It's therefore important for national programs to develop specific interventions for MSM in general and bisexual men in particular, to limit or even eliminate this spread of HIV within the couple and consequently in the general population.

HIV seroconversion is significantly associated with condom use in general and its use together with lubricant gel in particular. The risk of HIV infection was 3.8 times higher among MSM who reported not always using condoms plus lubricant during the three months preceding the survey compared to those who were still using it. Finally, MSM who did not consistently use condoms during anal sex with a male partner were 4 times more likely to be infected with HIV than others. These results are comparable to those of Guowu Liu et al who showed that having unprotected anal sex is a high-risk sexual behavior and can be considered as the main reason for HIV transmission among MSM [26]. Three essential factors can explain the non-use of condoms among MSM populations and expose them to HIV infection and other STIs, (i) complicity in the type of sexual relationship between the two persons or partners, which may lead them to believe that condoms interfere with their intimate relationship and are perceived as a barrier to their emotional development and their desire to give effectively themselves to each other [32, 40, 41], (ii) the level of HIV knowledge is reported to be closely related to the consistent use of condoms with casual or commercial male sexual partners [42]. A study by Dong et al shows that lower levels of knowledge about HIV and AIDS were important risk factors for HIV infection [22, 43]. The level of HIV knowledge among MSM in this study is quite low at the beginning of the monitoring, (iii) and concerning the availability and accessibility to these condoms [43, 44] despite the interventions received over the period in terms of HIV prevention packages (Fig 2). It's therefore important to implement interventions to increase and strengthen awareness-raising of MSM, especially young men, about knowledge, attitude and practice on HIV infection and access to condoms and lubricant gels.

## Limitations

The measurement of HIV infection event is realized only in the last month of the semester, which may lead to an underestimation of the real time of the occurrence of the event and

consequently of the number of people-time. This may result in an overestimation of the incidence. The results of our study will be difficult to generalize to the entire MSM population because it was conducted in major cities across the country with very little representativeness of northern populations and elderly people. But since the sampling method for the baseline source study is respondent- driven sampling and MSM identity associations have been called upon for the participation of all target categories, this risk may be reduced.

## Conclusion

The incidence of HIV infection is quite high among MSM populations, proving their ability to contribute to new HIV infections. The related socio-demographic and behavioral factors are also known. High-impact and cost-effective interventions still need to be developed and implemented to curb the epidemic in terms of universal access to prevention, care, support and treatment in an environment free of stigma and discrimination.

## Supporting information

**S1 File.**
(RAR)

## Acknowledgments

In the context of this research work, we would like to thank the offices of MSM association networks such as Benin Synergy Plus (BeSyP) and Coalition Sida Bénin (CSB) for sensitizing their peers to participate in the study and especially the follow-up over time during the 30 months. Our thanks also go to the health personnel who accompanied the clinical examination of the participants and the biological tests on a periodic basis.

## Author Contributions

**Conceptualization:** Septime P. H. Hessou, Yolaine Glele-Ahanhanzo, Michel Boko.

**Data curation:** Septime P. H. Hessou, Yolaine Glele-Ahanhanzo, Rhéda Adekpedjou, Clement Ahoussinou, Codjo D. Djade, Alphonse Biaou, Michel Alary.

**Formal analysis:** Septime P. H. Hessou, Yolaine Glele-Ahanhanzo, Rhéda Adekpedjou, Clement Ahoussinou, Alphonse Biaou, Michel Alary.

**Investigation:** Septime P. H. Hessou, Clement Ahoussinou.

**Methodology:** Septime P. H. Hessou, Yolaine Glele-Ahanhanzo, Clement Ahoussinou, Codjo D. Djade, Christian R. Johnson, Michel Boko.

**Project administration:** Septime P. H. Hessou.

**Resources:** Septime P. H. Hessou, Clement Ahoussinou.

**Software:** Septime P. H. Hessou.

**Supervision:** Septime P. H. Hessou, Rhéda Adekpedjou, Christian R. Johnson, Michel Boko, Michel Alary.

**Validation:** Septime P. H. Hessou, Christian R. Johnson, Michel Boko, Michel Alary.

**Visualization:** Septime P. H. Hessou.

**Writing – original draft:** Septime P. H. Hessou, Yolaine Glele-Ahanhanzo.

**Writing – review & editing:** Septime P. H. Hessou, Yolaine Glele-Ahanhanzo, Clement Ahoussinou, Codjo D. Djade, Christian R. Johnson, Michel Boko.

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
