## [Decision Letter · Decision Letter 0]

16 Mar 2020

PONE-D-19-36070

HIV Incidence and risk contributing factors among men who have sex with men in Benin: A prospective cohort study.

PLOS ONE

Dear Dr HESSOU,

Thank you for submitting your manuscript to PLOS ONE. After careful consideration, we feel that it has merit but does not fully meet PLOS ONE’s publication criteria as it currently stands. Therefore, we invite you to submit a revised version of the manuscript that addresses the points raised during the review process.

We would appreciate receiving your revised manuscript by Apr 30 2020 11:59PM. To enhance the reproducibility of your results, we recommend that if applicable you deposit your laboratory protocols in protocols.io, where a protocol can be assigned its own identifier (DOI) such that it can be cited independently in the future. For instructions see: http://journals.plos.org/plosone/s/submission-guidelines#loc-laboratory-protocols

We look forward to receiving your revised manuscript.

Kind regards,

Chiyu Zhang, Ph.D.

Academic Editor

PLOS ONE

Journal Requirements:

"NO"

Please provide an amended Funding Statement that declares *all* the funding or sources of support received during this specific study (whether external or internal to your organization) as detailed online in our guide for authors at http://journals.plos.org/plosone/s/submit-now.  Please state what role the funders took in the study.  If any authors received a salary from any of your funders, please state which authors and which funder. If the funders had no role, please state: "The funders had no role in study design, data collection and analysis, decision to publish, or preparation of the manuscript."

Reviewers' comments:

Reviewer's Responses to Questions

**Comments to the Author**

1. Is the manuscript technically sound, and do the data support the conclusions?

Reviewer #1: Yes

Reviewer #2: Yes

2. Has the statistical analysis been performed appropriately and rigorously? 

Reviewer #1: Yes

Reviewer #2: Yes

3. Have the authors made all data underlying the findings in their manuscript fully available?

Reviewer #1: Yes

Reviewer #2: Yes

4. Is the manuscript presented in an intelligible fashion and written in standard English?

Reviewer #1: Yes

Reviewer #2: Yes

5. Review Comments to the Author

Reviewer #1: This manuscript describes the HIV incidence among MSM in Benin. This prospective study in years revealed the quite high incidence of HIV infection among MSM populations, especially in <25y adult. It is of very significance in prevention and control of HIV in Africa countries , such as in Benin.

Reviewer #2: This manuscript by Septime P.H. Hessou et al reported HIV incidence and risk factors among MSM population in Benin bases on a prospective cohort study with a retention of more than 70%. Which is helpful to conduct intervention on MSM population. However, there are some items needed to further clear.

1. In table 1 and 3, occupation includes pupils and students. Could you explain that why pupils can sign consent form and also has behavior of sex with a man . It may be illegal to have sex with a pupil in any countries.

2. In result part, "the level of knowledge of getting HIV infected remained low at least during 18 months". Is this persistent low level of knowledge caused by no intervention measure taken, or no effect happened after intervention. If no intervention measure was taken, is it ethically permitted?

3. In this study, condom tearing reached to 27.8%. Could you explain why condom tearing so high. Is condom tearing related to no using lubricant or the quality of it. HIV infection would be prevented if without condom tearing.

4. There is some errors on the use of punctuation in table 4. For example, 0,92 should be 0.92. The same errors was also in context.

5. The author described that "The end of the follow-up takes place when the participant becomes HIV positive, that’s when he/she shows HIV positive". Is the female involved in this study?

6. PLOS authors have the option to publish the peer review history of their article (what does this mean?). If published, this will include your full peer review and any attached files.

Reviewer #1: No

Reviewer #2: Yes: Hongxiong Guo

---

## [Author Response · Author response to Decision Letter 0]

1 May 2020

PONE-D-19-36070

HIV Incidence and risk contributing factors among men who have sex with men in Benin: A prospective cohort study.

PLOS ONE

'Response to Reviewers'

Journal Requirements:

"NO"

a. Please provide an amended Funding Statement that declares *all* the funding or sources of support received during this specific study (whether external or internal to your organization) as detailed online in our guide for authors at http://journals.plos.org/plosone/s/submit-now. 

b. Please state what role the funders took in the study. If any authors received a salary from any of your funders, please state which authors and which funder. If the funders had no role, please state: "The funders had no role in study design, data collection and analysis, decision to publish, or preparation of the manuscript

Answer: This study was carried out as part of the research for my doctoral thesis.: "The funders had no role in study design, data collection and analysis, decision to publish, or preparation of the manuscript

5. Review Comments to the Author

Reviewer #1: This manuscript describes the HIV incidence among MSM in Benin. This prospective study in years revealed the quite high incidence of HIV infection among MSM populations, especially in <25y adult. It is of very significance in prevention and control of HIV in Africa countries, such as in Benin.

Answer: graded commentary

Reviewer #2: This manuscript by Septime P.H. Hessou et al reported HIV incidence and risk factors among MSM population in Benin bases on a prospective cohort study with a retention of more than 70%. Which is helpful to conduct intervention on MSM population. However, there are some items needed to further clear. 

1. In table 1 and 3, occupation includes pupils and students. Could you explain that why pupils can sign consent form and also has behavior of sex with a man . It may be illegal to have sex with a pupil in any countries.

Answer: As we can see in Tables 1 and 3, this category includes both pupils and students, and there is one essential notion that must be retained: the criteria for inclusion in the study: one must be MSM, but in addition one must be of legal age, i.e. at least 18 years old at the date of the survey. The conduct of this study has been approved by the National Committee for Ethics and Health Research in Benin (CNERS). The large number of subjects in this group (pupils/students) are students. It is by necessity of subject grouping and category of analysis that we have grouped the pupils/students together.

Thank you for your attention and understanding.

2. In result part, "the level of knowledge of getting HIV infected remained low at least during 18 months". Is this persistent low level of knowledge caused by no intervention measure taken, or no effect happened after intervention. If no intervention measure was taken, is it ethically permitted?

Answer: This is much more related to the lack of effect after the intervention or to an ongoing intervention. The preliminary results of this study have made it possible to correct and adapt intervention strategies in the course of implementation in relation to the specificity of the MSM target. It should be remembered that behavior change is also a process that lasts over time and that progressively gives results.

3. In this study, condom tearing reached to 27.8%. Could you explain why condom tearing so high. Is condom tearing related to no using lubricant or the quality of it. HIV infection would be prevented if without condom tearing.

Answer: The frequent tearing of condoms according to MSM would be related to the character and mechanism of the sexual act but especially and above all in the absence of the use of lubricating gels concomitantly with the condom. At the beginning of the study during the first six months of follow-up, just one-third of MSM (34.9%) used the condom coupled with the lubricating gel. The other issue is the availability and accessibility of free lubricant gel as well as condoms in intervention programs.

4. There is some errors on the use of punctuation in table 4. For example, 0,92 should be 0.92. The same errors was also in context.

Answer: remark taken into account and corrections made

5. The author described that "The end of the follow-up takes place when the participant becomes HIV positive, that’s when he/she shows HIV positive". Is the female involved in this study?

Answer: Cohort follow-up is conducted every 6 months with the team composed of the MSM informant (peer educator) for mobilization and confidence building of the MSM participating in the study, the laboratory technician for the HIV testing procedure and the physician in charge of the respondent's medical consultation and coordinator of the follow-up team. At the end of the follow-up, a global synthesis of the results is carried out and HIV-positive people are referred to the care site (testing and treat). Then, they leave the cohort follow-up as well as those lost to follow-up.

As indicated in the title and in the inclusion criteria, there are no female in the cohort.

---

## [Editor Report · Decision Letter 1]

11 May 2020

HIV Incidence and risk contributing factors among men who have sex with men in Benin: A prospective cohort study.

PONE-D-19-36070R1

Dear Dr. HESSOU,

We are pleased to inform you that your manuscript has been judged scientifically suitable for publication and will be formally accepted for publication once it complies with all outstanding technical requirements.

With kind regards,

Chiyu Zhang, Ph.D.

Academic Editor

PLOS ONE
---

## [Editor Report · Acceptance letter]

29 May 2020

PONE-D-19-36070R1 

HIV Incidence and risk contributing factors among men who have sex with men in Benin: A prospective cohort study. 

Dear Dr. Hessou:

I am pleased to inform you that your manuscript has been deemed suitable for publication in PLOS ONE. Congratulations! Your manuscript is now with our production department. 

With kind regards,

on behalf of

Dr. Chiyu Zhang 

Academic Editor

PLOS ONE